# VeChat: correcting errors in long reads using variation graphs

Xiao Luo[1,2], Xiongbin Kang[1] & Alexander Schönhuth ®[1,2] ✉

Error correction is the canonical first step in long-read sequencing data analysis. Current self-correction methods, however, are affected by consensus sequence induced biases that mask true variants in haplotypes of lower frequency showing in mixed samples. Unlike consensus sequence templates, graph-based reference systems are not affected by such biases, so do not mistakenly mask true variants as errors. We present VeChat, as an approach to implement this idea: VeChat is based on variation graphs, as a popular type of data structure for pangenome reference systems. Extensive benchmarking experiments demonstrate that long reads corrected by VeChat contain 4 to 15 (Pacific Biosciences) and 1 to 10 times (Oxford Nanopore Technologies) less errors than when being corrected by state of the art approaches. Further, using VeChat prior to long-read assembly significantly improves the haplotype awareness of the assemblies. VeChat is an easy-to-use open-source tool and publicly available at https://github.com/HaploKit/vechat.

Third-generation sequencing (TGS) such as single-molecule real-time (Pacific Biosciences, or short PacBio) or nanopore sequencing (Oxford Nanopore Technologies or short ONT) has been emerging rapidly over the last few years. The most obvious reason is that the length of TGS reads exceeds the length of next-generation sequencing (NGS) reads by orders of magnitude. While the length of TGS reads ranges from several Kbp up to even a few Mbp[1], NGS reads only span a few hundred base pairs. The fact that TGS is relatively inexpensive and, depending on the particular platform can even be carried out on mobile, handheld devices, greatly adds to its popularity. Last but not least, TGS does not suffer from PCR induced biases, because it circumvents polymerase chain reaction (PCR) as part of its protocol. Thanks to these advantages, TGS has been able to make decisive contributions in various areas of application. Prominent examples are haplotype phasing[2], genome assembly[3–7] and (complex) variant calling[8–10].

The downside of TGS, however, are the significantly elevated error rates the reads are subject to. For example, PacBio CLR and ONT reads, as the currently most representative examples of TGS reads, contain 5% to 15% errors[1]. This comes in obvious contrast to NGS short reads, whose error rates usually do not exceed 1%. The fact that the majority of errors affecting long reads consists of insertions and deletions adds to the difficulties because it prevents the application of principles and

straightforward adaptation of tools for correcting errors in short reads. This implies that direct usage of raw TGS reads or successful application of existing error correction tools is hardly possible in the majority of relevant applications. Novel methods and tools are required for correcting errors in TGS reads.

Because correcting errors in TGS reads is imperative for sound analyses, various TGS read error correction methods have been presented in the meantime. The corresponding range of methods can be divided into two major categories: hybrid correction and self-correction. While hybrid correction addresses to integrate short reads into the error correction process, self-correction seeks to correct errors without auxiliary data.

Hybrid correction reflects a sound and reasonable approach in general (see[11–14] for prominent approaches). However, hybrid correction suffers from certain pragmatic issues. First, while long reads can span repetitive regions, short reads cannot; this introduces ambiguities in the process of assigning short to long reads (or vice versa), and as a consequence biases in the quality of the correction, depending on the uniqueness of the region in the genome the reads stem from. Second, short reads re-introduce PCR induced biases. For example, certain areas of genomes are not sufficiently covered by short reads because of sequence content (e.g. GC content). This

[1]Genome Data Science, Faculty of Technology, Bielefeld University, Bielefeld, Germany. [2]Life Science & Health, Centrum Wiskunde & Informatica, Amsterdam, The Netherlands. ✉e-mail: a.schoenhuth@cwi.nl

hampers error correction in these areas. Last but not least, employing several different sequencing protocols can be impossible for equipment related or financial reasons, which prevents the application of hybrid error correction in the first place.

Self-correction, as the second class of methods, does not suffer from any of these issues. However, because of the lack of external (e.g. short read based) assistance, self-correction faces other methodically principled challenges. It is key to overcome these challenges before one can profit from the great practical advantages of self-error correction. In terms of prior, related work, self-correction can be further divided into three sub-categories, each of which is characterized by particular algorithmic strategies and methodical foundations.

The first, and most common of the three categories is based on multiple sequence alignments (MSAs). For prior approaches and tools that crucially rely on computing MSAs, see Racon[15], the error correction module of the assembler Canu[16], and FLAS[17]. The second principled class of approaches relies on de Bruijn graphs (DBGs). Corresponding tools employ DBGs at some point crucial for the correction process. The prevalent tool to consider is Daccord[18], which is based on raising local DBGs, where local refers to reads, from which DBGs are constructed, stemming from relatively small segments of the genome. The third class of self-correction methods collects approaches that make combined use of both MSAs and DBGs. Such methodically combined approaches seek to balance the advantages and disadvantages of the two concepts, MSAs on the one hand, and DBGs on the other hand. Prominent tools that make successful, combined use of MSAs with DBGs are LoRMA[19] and CONSENT[20].

The common denominator that unifies all of these self-correction approaches is to raise consensus sequence as a summary of the reads observed. This consensus sequence then serves as a template during error correction, by indicating default variation. However, because sequence-based templates cannot capture ambiguities, one experiences biases during the correction process: the default allele provided by the template wins in case of uncertainties remaining. As a consequence, these approaches tend to mask variation that characterizes little-covered or low-frequency haplotypes/strains in mixed samples (metagenomes, cancer genomes) or polyploid genomes. Haplotypes exhibiting template masked variants virtually disappear, such that downstream analyses remain blind to them.

To address this issue, we suggest VeChat ([V]ariation graph-based [e]rror [C]orrection in [ha]plo[t]ypes), a self-correction method to perform haplotype-aware error correction for long reads. From a larger perspective, VeChat considers the full spectrum of all possible haplotypes that possibly affect the sample already during error correction, and not−as is common−only thereafter. This reflects a novelty for ploidies larger than two, because earlier approaches only deal with diploid scenarios[21]. From a methodical point of view, the novelty of VeChat is to integrate variation graphs[22] as a fundamental data structure into the process of error correction. Variation graphs have been effectively used to solve various problems in computational genomics, such as improving read mapping and variant calling[23–25], modeling haplotypes[26] and assembling genomes from mixed samples[27,28]. To the best of our knowledge, VeChat is the first approach to apply variation graphs to long-read error correction. We have tested VeChat and extensively compared it with the current state of the art on datasets reflecting various settings of current interest. Benchmarking experiments on both simulated and real data demonstrate that our approach basically achieves the best performance rates, across all categories of common sequencing errors. Moreover, using VeChat for long-read error correction prior to haplotype-aware genome assembly largely improves assemblies in terms of most relevant categories, most prominently including completeness, contiguity and accuracy.

## Results

We have designed and implemented VeChat, an approach to haplotype aware long-read self error correction. The key concept underlying VeChat are variation graphs. Unlike single consensus sequences, which current self-correction approaches are generally centering on, variation graphs are able to represent the genetic diversity across multiple, evolutionarily or environmentally coherent genomes[22]. This enables to preserve haplotype-specific variation during error correction also for samples of higher, known or unknown ploidy.

In this section, we first provide a high-level description of the workflow of VeChat. We then evaluate the performance of VeChat on both simulated and real data in comparison with the state of the art approaches. Finally, we assess the effect of integrating VeChat as a preprocessing tool in common haplotype aware genome assembly pipelines.

### Workflow

See Fig. 1 for an illustration of the workflow of VeChat. See also Methods for full details in the following.

The basic idea of VeChat is to construct a variation graph from the all-to-all alignments of the raw reads. One then identifies nodes and edges in the resulting graph that are likely to be artifacts, and removes them. Subsequently, reads are realigned against the resulting, pruned graph. The path in the pruned graph corresponding to the optimal realignment points out an error-corrected sequence of the read. The procedure of spurious node and edge removal followed by realignment is repeated until convergence (note that the statistical evaluation of remaining nodes and edges changes upon re-alignment, which may reveal new likely spurious nodes and edges in the next iteration). The re-alignment of the original read with the final graph points out the fully error-corrected sequence of the read.

VeChat consists of two cycles. While the first cycle yields precorrected reads, the second cycle generates the final, corrected reads from the pre-corrected reads. Each cycle proceeds in 6 steps. While the two cycles generally agree on these 6 steps, they disagree in terms of small, but crucial details affecting steps 1 and 4.

During the first cycle, step 1 computes minimizer based all-vs-all overlaps, for which we employ Minimap2[29]. Minimap2 prevents the need for computing base-level alignments. Therefore, this stage proceeds rapidly and without additional efforts.

Steps 2–6 reflect the technical core of the error correction procedure in Fig. 1. In step 2, a target read is selected as the read whose errors are to be corrected. A read alignment pile that consists of all reads that overlap it is computed. Subsequently, in step 3, the read alignment pile is divided into small segments, where each of the segments gives rise to a window like part of the pile in step 3; the part of the target read in a particular window is further referred to as 'target subread'.

Subsequently, in step 4, the error correction for target subreads is performed in each window. Step 4 is methodically more involved, because it captures the novel, variation graph-based approach; see Fig. 2 for detailed illustrations on the version of that particular step used in the first cycle. Step 4 involves the construction of a variation graph using the partial order alignment (POA) algorithm[30], and pruning this graph in an iterative manner from nodes and edges that are spurious because they reflect errors ('Graph pruning' and 'Graph re-pruning' in Fig. 2). For pruning the graph, we make use of a frequent itemset model that involves read coverage, sequencing errors and co-occurrence of characters in reads. The path in the pruned variation graph that corresponds to the optimal alignment of the target subread is then taken as the pre-corrected target subread; see subsection Step 4: Error correction for target subreads in Methods for details. The first cycle concludes with concatenating the different 'target subreads' of one target read, which results in a pre-corrected read at full, original

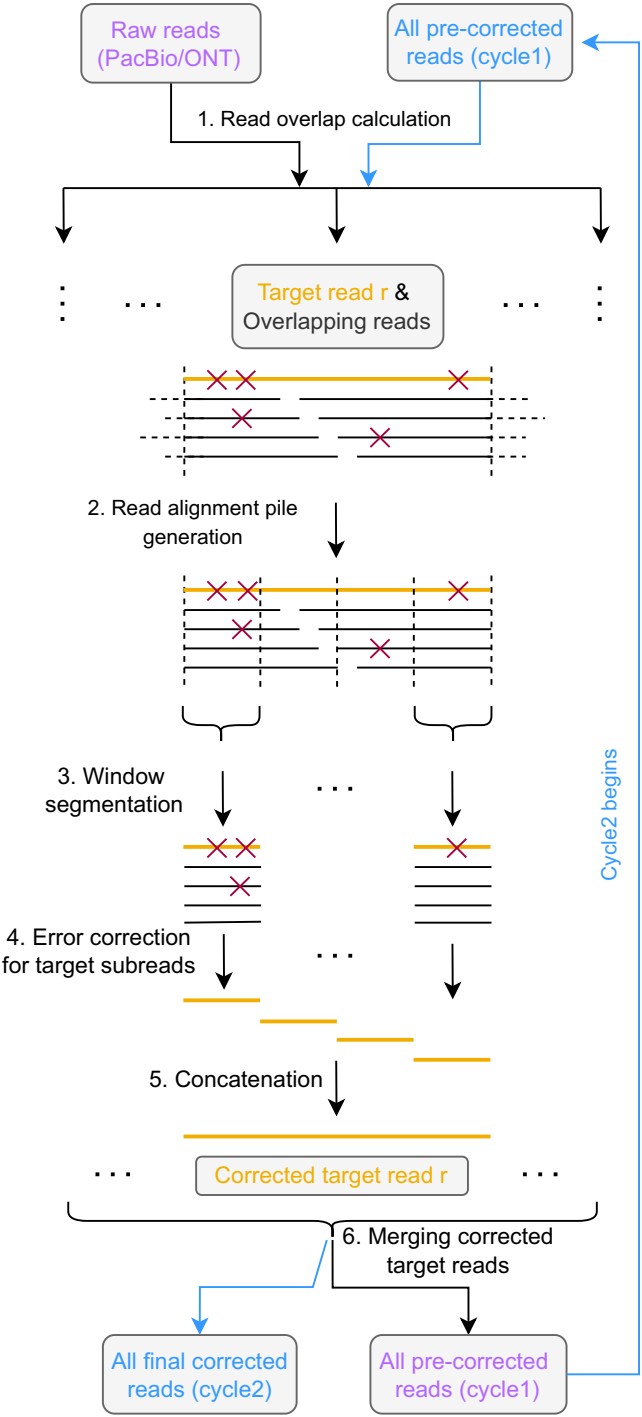

**Fig. 1 | Workflow of VeChat.** The input and output of cycle 1 and cycle 2 are labeled with purple and blue, respectively. Both cycle 1 and cycle 2 share the steps 1-6 except some differences in step 1 and 4. The target read is highlighted with orange. Red forks indicate the sequencing errors in reads.

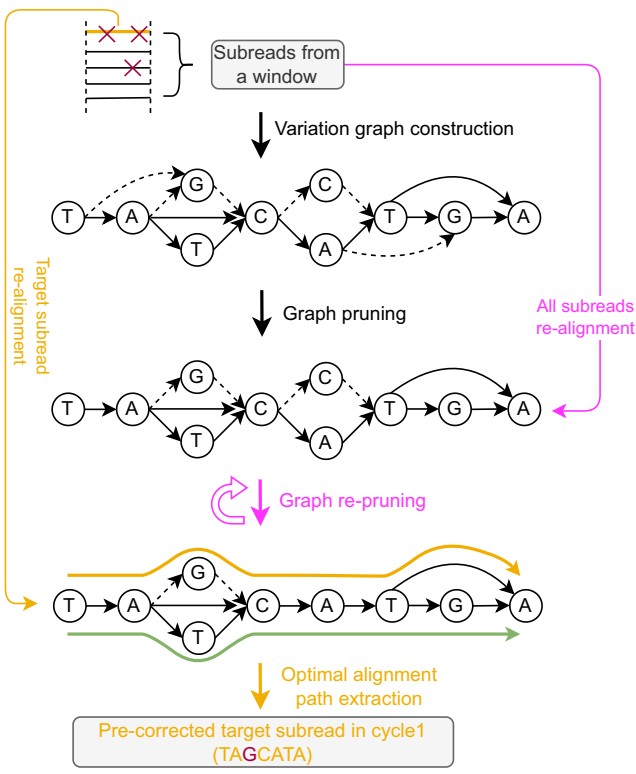

**Fig. 2 | Error correction for one target subread in cycle 1.** The error correction process for the target read *r* is illustrated assuming a diploid scenario (the orange path represents the optimal alignment path, whereas the green path represents the other true haplotype). "Graph pruning" and "Graph re-pruning" refer to the core error correction procedures. These procedures rely on a variation graph that is constructed from segments of a read alignment pile that results from a multiple alignment of the target read and the reads that overlap it, see Fig. 1. During graph pruning and re-pruning spurious edges (dashed arrows), induced by sequencing errors, are removed from the variation graph. The pink elements indicate that these procedures are repeated.

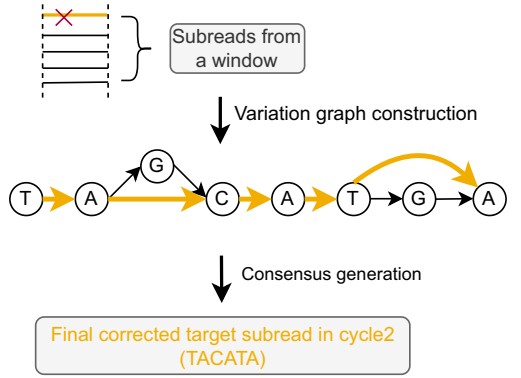

**Fig. 3 | Error correction for one target subread in cycle 2.** The bold orange path indicates the consensus sequence. The false nucleotide 'G' in the pre-corrected target subread in cycle 1 (see in Fig. 2) is marked with red and further corrected in cycle 2.

length. These pre-corrected reads then serve as input for the second cycle.

While the (vast) majority of errors have already been corrected during the first cycle, a few errors, to be considered statistical outliers that escape correction during iterative graph pruning during the first cycle, have resisted their correction. The second cycle is supposed to spot such outliers. The second cycle is less complex than the first cycle, because it does no longer include the statistically involved graph pruning procedure; the blue elements in Fig. 1 point out the different routes along which the second cycle proceeds. Overall, as above-

mentioned, the second cycle is identical with the first cycle in steps 2, 3, 5 and 6. In step 1, however, beyond computing all-vs-all overlaps, base-level alignments are computed, which enables haplotype aware read overlap filtration. Step 4, as shown in Fig. 3, then proceeds differently insofar as graph (re-)pruning is no longer part of the cycle. Instead of iterative re-pruning, which did not lead to removal of the errors that we would like to remove during this second cycle, (haplotype-aware!) consensus sequences (displayed as thick yellow arrows in

**Table 1 | Error correction benchmarking results for simulated PacBio CLR reads of various polyploid genomes (ploidy = 2, 3, 4)**

| Method | #Reads | Error rate (%) | Mismatch (%) | Indel (%) | Haplotype coverage (%) | N50 (bp) | NGA50 (bp) | #Misassemblies |
|---|---|---|---|---|---|---|---|---|
| **Ploidy = 2** | | | | | | | | |
| VeChat | 31958 | 0.014 | 0.006 | 0.008 | 100.0 | 12,556 | 38,515 | 0 |
| CONSENT | 33115 | 0.194 | 0.152 | 0.042 | 99.9 | 12,509 | 38,318 | 38 |
| Racon | 32183 | 0.276 | 0.190 | 0.085 | 99.2 | 12,514 | 38,421 | 72 |
| Canu | 25924 | 0.308 | 0.183 | 0.124 | 99.9 | 13,280 | 38,517 | 3 |
| Daccord | 31403 | 0.423 | 0.412 | 0.011 | 99.2 | 12,604 | 38,598 | 3 |
| **Ploidy = 3** | | | | | | | | |
| VeChat | 48085 | 0.031 | 0.015 | 0.016 | 100.0 | 12,595 | 38,467 | 13 |
| CONSENT | 50462 | 0.276 | 0.205 | 0.071 | 100.0 | 12,511 | 38,187 | 105 |
| Racon | 48986 | 0.558 | 0.427 | 0.131 | 98.7 | 12,484 | 38,257 | 288 |
| Canu | 37210 | 0.612 | 0.405 | 0.207 | 99.9 | 13,675 | 38,360 | 30 |
| Daccord | 48189 | 0.807 | 0.752 | 0.055 | 99.7 | 12,485 | 38,357 | 16 |
| **Ploidy = 4** | | | | | | | | |
| VeChat | 62743 | 0.074 | 0.047 | 0.027 | 99.9 | 12,593 | 38,442 | 44 |
| CONSENT | 66315 | 0.275 | 0.180 | 0.095 | 100.0 | 12,492 | 38,268 | 111 |
| Racon | 64342 | 0.547 | 0.398 | 0.149 | 93.2 | 12,464 | 38,016 | 387 |
| Canu | 46698 | 0.549 | 0.335 | 0.214 | 99.5 | 13,956 | 38,255 | 86 |
| Daccord | 63440 | 0.833 | 0.790 | 0.043 | 97.5 | 12,463 | 38,335 | 22 |

The average sequencing coverage per haplotype is 30× and sequencing error rate is 10%. '#Reads' indicates the number of corrected reads. The error rate is equal to the sum of mismatch and indel rate. The results are sorted by the error rate in ascending order.

Fig. 3) are derived from the constructed variation graphs by using a dynamic programming algorithm[31]. After concatenating these consensus sequences in step 5, joining all target reads in step 6 generates the final output of Vechat.

**Benchmarking results**

Table 1 shows the error correction benchmarking results for simulated PacBio CLR reads from genomes of varying ploidies, namely 2,3 and 4. VeChat achieves approximately 14–30, 9–26 and 4–11 times lower error rates on diploid, triploid and tetraploid genomes, respectively. At the same time, it maintains better or comparable performance in terms of other aspects such as number of corrected reads, completeness (haplotype coverage), number of misassemblies and length of corrected reads (as shown by N50/NGA50). In particular, VeChat outperforms other tools in terms of mismatch rate (4–69 times lower than others).

Table 2 shows the error correction benchmarking results for simulated Oxford Nanopore reads from genomes of varying ploidies, namely 2, 3 and 4. VeChat achieves approximately 10–20, 3–9 and 2–5 times lower error rates on diploid, triploid and tetraploid genomes, respectively, while maintaining better or comparable performance in terms of all other aspects. Just as for PacBio reads, VeChat also shows better performance in terms of mismatch rate: 2–59 times lower than other correction tools, compared with indel rate.

Table 3 shows the error correction benchmarking results for simulated PacBio CLR reads of metagenomic datasets with different complexity. VeChat achieves approximately 6–7 and 3–4 times lower error rates on low and high complexity metagenomes, respectively, while maintaining comparable performance in terms of other aspects. In particular, VeChat outperforms other tools in terms of mismatch rate (6–12 times lower) quite substantially on the low-complexity dataset.

Table 4 shows the error correction benchmarking results for real PacBio sequencing data (mock communities). The three sections of the table show results on the yeast pseudo-diploid genome dataset (mock community) first, the NWC metagenome dataset (real) second, and the Microbial 10-plex metagenome dataset (mock community) as the third section of rows in Table 4. VeChat achieves approximately 2–4, 1.4–7.8

and 3.3–5.6 times lower error rates on Yeast, NWC and Microbial 10-plex datasets, respectively, while maintaining comparable performance in terms of other aspects.

See Supplementary Table 1 for error correction benchmarking experiments for real ONT sequencing data (non-synthetic), which have been evaluated using Merqury[32] because of the lack of reference genomes. Before discussing results, see Supplementary Table 2, which puts evaluations with and without a reference genome (QUAST resp. Merqury), that is, with and without available ground truth into context. Corresponding results immediately point out that Merqury is subject to substantial biases with respect to the choice of methods. For example, on ONT sequenced diploid genomes, Merqury underestimates the true error rates (as performed by QUAST relative to the ground truth) by factors of 4.33 (CONSENT) and even 11.84(!) (Daccord), but only by factors of 2.54 (Racon), 2.05 (Canu), and 1.75 (VeChat).

The quality of the results persists on the other datasets: Merqury evidently favors CONSENT and Daccord quite substantially in comparison with Racon, Canu and VeChat. Because Merqury is k-mer based, an immediate hypothesis is that Merqury tends to favor k-mer (e.g. de Bruijn graph) based approaches (CONSENT, Daccord) over approaches that do not make use of de Bruijn graphs (Racon, Canu and VeChat), where VeChat appears to be the only tool whose error rates are not substantially underestimated, at least on the lesser complex datasets. In summary, we have experienced that Merqury is affected by considerable volatility with respect to the methodological background of error correction tools, clearly favoring certain tools over others.

Therefore, the discussion of the following results are to be taken with the corresponding caution in terms of the method-specific biases that Merqury appears to induce.

As becomes obvious from Supplementary Table 1, VeChat achieves approximately 1.5 times lower error rate (QV) and 1.2 times lower switch error on HG002 compared with CONSENT (the only alternative tool available to compare), while loosing more haplotype coverage. Whereas on the human gut microbiome dataset, Daccord achieves the lowest error rate (QV) while VeChat obtains comparable read accuracy. VeChat achieves about 1.4 and 1.7 times lower error rates (QV) in comparison to CONSENT and Canu, respectively, while

**Table 2 | Error correction benchmarking results for simulated Oxford Nanopore reads of various polyploid genomes (ploidy = 2, 3, 4)**

| Method | #Reads | Error rate (%) | Mismatch (%) | Indel (%) | Haplotype coverage (%) | N50 (bp) | NGA50 (bp) | #Misassemblies |
|---|---|---|---|---|---|---|---|---|
| **Ploidy = 2** | | | | | | | | |
| VeChat | 30920 | 0.022 | 0.007 | 0.014 | 99.9 | 13,095 | 40,612 | 5 |
| CONSENT | 32661 | 0.212 | 0.160 | 0.052 | 99.9 | 13,040 | 40,903 | 37 |
| Racon | 31840 | 0.346 | 0.234 | 0.112 | 99.3 | 13,039 | 40,725 | 120 |
| Canu | 25506 | 0.390 | 0.206 | 0.183 | 100.0 | 13,820 | 40,846 | 6 |
| Daccord | 31438 | 0.438 | 0.410 | 0.027 | 99.2 | 12,987 | 40,730 | 3 |
| **Ploidy = 3** | | | | | | | | |
| VeChat | 45113 | 0.090 | 0.041 | 0.050 | 100.0 | 13,130 | 40,497 | 90 |
| CONSENT | 49826 | 0.298 | 0.221 | 0.077 | 99.9 | 13,037 | 40,877 | 103 |
| Racon | 48520 | 0.673 | 0.501 | 0.172 | 98.6 | 13,028 | 39,286 | 631 |
| Canu | 36962 | 0.748 | 0.453 | 0.295 | 99.9 | 14,141 | 40,605 | 59 |
| Daccord | 49033 | 0.821 | 0.751 | 0.069 | 99.7 | 12,712 | 39,193 | 11 |
| **Ploidy = 4** | | | | | | | | |
| VeChat | 58739 | 0.169 | 0.098 | 0.071 | 99.7 | 13,129 | 40,450 | 177 |
| CONSENT | 65384 | 0.292 | 0.195 | 0.097 | 100.0 | 13,033 | 40,864 | 121 |
| Racon | 63670 | 0.666 | 0.469 | 0.197 | 94.9 | 13,012 | 40,007 | 668 |
| Daccord | 65072 | 0.840 | 0.784 | 0.056 | 96.9 | 12,606 | 39,076 | 22 |

The average sequencing coverage per haplotype is 30× and sequencing error rate is 10%.

**Table 3 | Error correction benchmarking results for simulated PacBio CLR reads of metagenomic datasets with different complexity**

| Method | #Reads | Error rate (%) | Mismatch (%) | Indel (%) | Haplotype coverage (%) | N50 (bp) | NGA50 (bp) | #Misassemblies |
|---|---|---|---|---|---|---|---|---|
| **Low complexity (20 genomes)** | | | | | | | | |
| VeChat | 293466 | 0.036 | 0.020 | 0.015 | 96.9 | 11,866 | 29,555 | 104 |
| Racon | 299053 | 0.200 | 0.122 | 0.078 | 91.7 | 11,811 | 29,514 | 794 |
| CONSENT | 299333 | 0.214 | 0.149 | 0.065 | 98.4 | 11,841 | 29,556 | 515 |
| Canu | 253381 | 0.259 | 0.134 | 0.125 | 97.4 | 12,370 | 29,457 | 139 |
| Daccord | 298284 | 0.259 | 0.243 | 0.016 | 92.8 | 11,862 | 29,595 | 280 |
| **High complexity (100 genomes)** | | | | | | | | |
| VeChat | 1441190 | 0.088 | 0.061 | 0.026 | 97.5 | 11,886 | 30,129 | 2774 |
| CONSENT | 1497216 | 0.274 | 0.163 | 0.112 | 99.4 | 11,839 | 30,204 | 3263 |
| Canu | 1185152 | 0.354 | 0.192 | 0.162 | 99.0 | 12,706 | 30,016 | 873 |
| Racon | – | – | – | – | – | – | – | – |
| Daccord | – | – | – | – | – | – | – | – |

The average sequencing coverage of strains is about 30x and the sequencing error rate is 10%. Racon and Daccord failed to run for high complexity dataset.

keeping comparable performance in terms of other aspects. (We recall that Daccord was the tool whose error rate was underestimated by the by far largest factors, which points out that, potentially, VeChat virtually achieves better error rates). As we mentioned earlier in the subsection 'Metrics for evaluation', the error rate (QV) ignores long-range switch errors in evaluation, which is unable to represent the overall error rate of reads. In fact, in simulated datasets of which the ground truth are known, we observed that VeChat achieves much lower switch error rate compared with others, and in both metagenome datasets VeChat achieves much lower overall error rate (from QUAST), even though its error rate (QV, from Merqury) is comparable with Daccord in the metagenome dataset of low complexity (see Supplementary Table 2). In summary, we speculate VeChat can achieve better performance in terms of overall error rate on the real human gut microbiome data.

**Varying read coverage**

In order to evaluate how sequencing coverage influences the error correction approaches, we focused on the triploid genome, consisting of three E. coli strains as described before. We simulated PacBio CLR reads at varying sequencing coverage, namely, 10×, 20×, 30×, 40×, 50× per haplotype.

Supplementary Table 3 shows the benchmarking results of error correction. In summary, VeChat achieves approximately 2–47 times lower error rates on all datasets, while maintaining better or comparable performance in terms of other aspects such as number of corrected reads, completeness (HC) and length of corrected reads. As the sequencing coverage increases (from 10× to 50×), VeChat achieves better error correction (error rate from 0.311% to 0.017%), while keeping comparable performance in terms of other aspects.

In addition, we particularly tested VeChat in the scenario of ultra-high sequencing coverage over a small genome. To reflect this context, we simulated a 5-strain HIV mixture (genome size ≈ 10 Kbp) dataset, which has been used for benchmarking experiments in many related studies, such as refs. 27, 33–35. The average sequencing coverage per strain is about 1000×. See the Supplementary Table 4 for the details about the data descriptions and the benchmarking results. The results show that VeChat outperforms others on the PacBio data in terms of

**Table 4 | Error correction benchmarking results for real PacBio sequencing data (mock communities)**

| Method | #Reads | Error rate (%) | Mismatch (%) | Indel (%) | Haplotype coverage (%) | N50 (bp) | NGA50 (bp) | #Misassemblies |
|---|---|---|---|---|---|---|---|---|
| **Yeast pseudo-diploid genome** | | | | | | | | |
| VeChat | 107210 | 0.236 | 0.111 | 0.126 | 99.6 | 5693 | 15,537 | 505 |
| Daccord | 149020 | 0.503 | 0.285 | 0.217 | 96.5 | 4762 | 15,161 | 1457 |
| Racon | 136199 | 0.758 | 0.282 | 0.476 | 98.3 | 6349 | 16,001 | 3836 |
| Canu | 118367 | 0.787 | 0.214 | 0.573 | 99.9 | 5684 | 15,603 | 743 |
| CONSENT | 160136 | 0.947 | 0.344 | 0.603 | 99.4 | 5622 | 16,001 | 7973 |
| **NWC metagenome** | | | | | | | | |
| VeChat | 156426 | 0.101 | 0.031 | 0.070 | 99.3 | 9619 | 27,416 | 14961 |
| Daccord | 163313 | 0.140 | 0.079 | 0.061 | 99.5 | 8461 | 26,343 | 12440 |
| Racon | 168879 | 0.394 | 0.062 | 0.332 | 99.5 | 9914 | 28,428 | 10832 |
| Canu | 37779 | 0.551 | 0.090 | 0.461 | 99.1 | 13811 | 27,729 | 4066 |
| CONSENT | 176764 | 0.787 | 0.107 | 0.680 | 99.7 | 9708 | 27,638 | 9731 |
| **Microbial 10-plex metagenome** | | | | | | | | |
| VeChat | 245804 | 0.089 | 0.066 | 0.023 | 99.3 | 7837 | 17,511 | 1533 |
| Racon | 253817 | 0.297 | 0.160 | 0.137 | 97.8 | 8019 | 17,760 | 3724 |
| Canu | 193810 | 0.328 | 0.121 | 0.206 | 99.8 | 8477 | 17,824 | 1170 |
| Daccord | 254003 | 0.336 | 0.298 | 0.038 | 98.2 | 7704 | 17,342 | 2073 |
| CONSENT | 256935 | 0.495 | 0.107 | 0.388 | 100.0 | 8017 | 17,729 | 3470 |

error rate, whereas it achieves the second best performance on the Nanopore data. In summary, VeChat still works for the ultra-high sequencing coverage case, but is not necessarily very effective especially for ONT reads.

### Varying sequencing error rates

In order to evaluate the effect of sequencing error rate on the different methods, we again focused on the triploid genome consisting of three E. coli strains as described above. Accordingly, we simulated PacBio CLR reads at varying sequencing error rates, namely, at 5%, 10% and 15%. The average sequencing coverage per haplotype is 30x.

Supplementary Table 5 shows the corresponding benchmarking results of error correction. Overall, VeChat achieves approximately 10–93, 9–26 and 7–9 times lower error rates on datasets with 5%, 10% and 15% errors, respectively, while maintaining comparable performance in terms of other aspects. On decreasing sequencing error rate (from 15% down to 5%), VeChat's error correction undoubtedly improves with error rates dropping from 0.091% to 0.009%.

### Improving genome assembly

De novo genome assembly sticks out among the many possible applications that depend on sequencing reads in two ways. First, sequencing reads are its only input. Second, genome assembly seeks to reconstruct the very sequences from which the reads stem. In other words, arguably, de novo assembly is the primary application of genome sequencing.

It is therefore also of primary interest how correcting errors in reads influences the quality of de novo genome assemblies that build on the corrected reads.

To investigate related effects, we carried out some straightforward experiments, and ran (Hi)Canu[16,36] and (meta)Flye[6,37], as the predominant long-read genome assemblers, with and without correcting reads using VeChat before. We did this on the datasets described earlier.

Before discussing results, note that the construction of pipelines that are optimized in terms of combining error correction and genome assembly approaches requires further investigation, which is beyond the scope of this manuscript. Here, we report the quick application "VeChat + Assembler" on various polyploid genome and metagenome datasets.

See Supplementary Tables 6–9 for details on the following results. We evaluated genome assemblies in terms of various relevant aspects, again using Quast.

It becomes immediately evident that (Hi)Canu and (meta)Flye profit from VeChat corrected reads considerably. On the vast majority of datasets, greater haplotype coverage, longer contigs (N50/NGA50), lower error rates (especially mismatch rates) and fewer misassemblies are reported upon prior application of VeChat.

For example, in the diploid case (Supplementary Table 6), VeChat +HiCanu increases haplotype coverage (HC) by 7.6% over Canu, from 92% to 99.6%, which also implies that VeChat+HiCanu supports to reconstruct both haplotypes at nearly full length. N50/NGA50 is about 10 times greater for VeChat+HiCanu. Also, VeChat+HiCanu yield fewer number of misassemblies, and error rate is more than 90 times lower. The substantial improvements of VeChat+HiCanu persist when raising the ploidy. We note particularly that VeChat+HiCanu increase the haplotype coverage by 14% in tetraploid genomes, at remarkably enhanced alternative metrics as well. Moreover, VeChat+Flye substantially improves genome assemblies for the triploid and the tetraploid case, compared with Flye alone. Overall, this points out that one can use VeChat for substantially improving on polyploid genome assembly quality when working with state-of-the-art assemblers for long reads.

### Runtime and memory usage evaluation

The runtime of VeChat is dominated by three steps: while the computation of read overlaps (without base-level alignment) is fast, subsequent edit-distance-based alignment (for segmenting windows) is time-consuming. Second, the POA algorithm that drives the construction of the variation graphs performs sequence-to-graph alignment, which comes at computational complexity of $O(N(2N_p + 1)|V|)$, where $N$ is the length of the sequence to be aligned, $N_p$ is the average number of predecessors in the graph and $|V|$ is the number of vertices in the graph[30]. Third, VeChat follows an iterative paradigm, such as read overlap computation (with base-level alignment) and error correction (consensus generation) steps during the second iteration; this also requires a non-negligible amount of running time.

We performed most of benchmarking analyses on x86_64 GNU/Linux machines using 48 cores. The runtime and peak memory usage evaluations for different methods are reported in Supplementary Tables 10–13. VeChat takes 23–81 CPU hours and 8–92 CPU hours on

simulated PacBio CLR and ONT reads from datasets reflecting varying ploidy (2, 3 or 4, as usual), which is 1.1–6.2 and 0.6–6.9 times slower than other methods. At the same time, it requires higher peak memory usages (Supplementary Tables 10 and 11). VeChat is 2.4–7.1 times slower than other approaches on the simulated metagenomic dataset, and reaches higher peak memory usages (Supplementary Table 12). Note that the high complexity metagenomic dataset is too large to be processed by Racon and Daccord, whereas our approach is able to handle it. In addition, VeChat is 0.7 ~ 32 times slower on real sequencing datasets (mock communities), while requiring higher peak memory usages (Supplementary Table 13).

## Discussion

We have presented VeChat, as an approach that performs haplotype-aware error correction for third-generation sequencing (TGS) reads. To the best of our knowledge, VeChat is the first approach that explicitly addresses to preserve haplotype-specific variation already during the correction process. The methodical improvement over prior approaches has been to make use of graph-based instead of sequence-based reference systems, which avoids the typical consensus sequence-induced biases.

Results have demonstrated the superiority of VeChat: in all benchmarking scenarios, VeChat suppresses error rates by at least a factor of 1–3, if not, as is the case for the majority of scenarios, suppressing error rates by one or even two orders of magnitude in comparison with the leading competitive approaches. At the same time, VeChat preserves the haplotype identity of the reads, which means that after correction with VeChat, all reads contribute to the coverage of the haplotype they stemmed from. The most obvious interpretation of these results is that capturing haplotype structure already during error correction is not just beneficial, but perhaps even imperative, when seeking to remove all errors from TGS reads.

For appropriately capturing haplotype-specific variants during the error correction step, we construct variation graphs from the noisy TGS reads directly. Note that direct construction of variation graphs from heavily erroneous reads is not standard. In fact, at first glance, it is even counterintuitive, because the seminal idea of variation graphs is to be constructed from true haplotype-specific, sufficiently long patches of sequence. Here, patches of sequence contain up to even 15% of errors.

As a consequence, upon initial construction, the graphs contain a large amount of nodes and edges that reflect sequencing errors. For identifying spurious nodes and edges, one exploits that sequencing errors are randomly distributed, whereas variants tend to re-occur across different reads. In particular, edges that link spurious nodes (with true nodes or other spurious nodes) tend to be little covered by reads, because reads do not tend to share errors, whereas they do tend to share true variation. To systematically identify spurious edges as edges that are covered by too little reads, in a sound, principled way, we adopt two metrics from frequent itemset mining. While Support measures the relative coverage of edges in the graph, Confidence measures the association between the two nodes incident to the edge they share; if basic support or the association is too little, the edge and possibly resulting isolated nodes are removed.

A particular effect of VeChat is to achieve substantial improvements in terms of mismatch rates; improvements on indel rates are also evident in all scenarios, but usually a bit less substantial. One possible explanation is that substitution events, much more than insertion and deletion events, dominate the evolutionary processes of living organisms, and thus are often characteristic of strains or haplotypes. VeChat appears to be the first approach to correctly preserve these single nucleotide polymorphisms (SNPs), because the distinction of haplotypes is just what variation graphs are made for. At any rate, VeChat appears to prevent masking of true variants as a consequence of generating consensus sequence.

Further, we have demonstrated that correcting reads using VeChat prior to performing de novo genome assembly significantly enhanced the resulting assemblies. Obviously, and unsurprisingly, not mistaking haplotype specific variation as errors, and so preserving the haplotype identity of the reads results in assemblies that are considerably enhanced in terms of haplotype awareness. Particularly notably, correcting reads with VeChat enabled the application of HiCanu thereafter, as the best strategy evaluated here, which is interesting because HiCanu crucially relies on clean reads. Nevertheless, the exploration of optimal strategies and pipelines in terms of combining preprocessing, error correction and core assembly tools, relative to particular settings such as polyploid, cancer or metagenomes, still requires further efforts, which are beyond the scope of this study.

As for future perspectives, the rapid development of long-read sequencing technologies will lead to decreasing sequencing error rates. However, because the advantages of VeChat become even more evident when sequencing error rates drop, VeChat will also be a superior tool when correcting long reads in which errors appear at a rate of 5% or lower (see Supplementary Table 5).

Of course, future improvements are conceivable: in particular, although not requiring excessive amounts, VeChat does not win the competition in terms of computational resources. In particular on large datasets, VeChat experiences longer runtimes and higher peak memory usage. However, there is room for improving on that point. VeChat uses off-the-shelf approaches in some places, without making use of all features these off-the-shelf approaches provide. This amounts to unnecessary overheads when running these tools, which one can avoid by disintegrating approaches into their single functionalities and running them separately. In particular, there is good hope that computations such as edit-distance-based alignments, or the sequence-to-graph alignments, can be replaced by more efficient routines in the future.

## Methods

### Datasets

We made use of PBSIM2[38], as a most recent tool to simulate PacBio CLR and Oxford Nanopore reads using built-in `P6C4` and `R103` model-based simulation profiles, respectively. Since the main application scenario of VeChat is to correct long-read sequencing data from multiple genomes, such as polyploid genomes and metagenomes, we simulated various datasets for both cases.

We constructed pseudo diploid (ploidy = 2, ANI: 98%), triploid (ploidy = 3, ANI: 96–98%) and tetraploid (ploidy = 4, ANI: 96–99%) genomes by mixing strains of *Escherichia coli* (*E. coli*) bacteria; note that Average Nucleotide Identity (ANI) is defined to measure the genome sequence similarity, which can be reported by FastANI[39], for example. All genome sequences of E. coli were downloaded from the NCBI database (see Supplementary Data 1 for the details of reference genomes). Reads were simulated from the haplotypes (i.e. strains) independently and upon generation mixed together to form the corresponding polyploid genome datasets (ploidy = 2,3,4). We simulated both PacBio CLR and Nanopore reads for these datasets, at average sequencing coverage of 30x per haplotype and average sequencing error rate of 10%.

Additionally, we used CAMISIM[40] to simulate two metagenomic datasets (PacBio CLR reads) of different levels of complexity. Here, we used PBSIM2 to simulate PacBio CLR reads instead of the built-in simulator in CAMISIM. The low-complexity dataset consists of 10 species (20 strains), whereas the high complexity dataset consists of 30 species (100 strains). The genomes used in both datasets are derived from[41] (see Supplementary Data 1 for the details). For both datasets, the average sequencing coverage of strains is about 30x and the average sequencing error rate is 10%. The relative abundances of strains range from 1.9% to 10.6% and from 0.28% to 3.3%, respectively. The corresponding sequencing coverages over different strains range

from 12.6× to 68.0× and from 11.0× to 126.9×, respectively; see Supplementary Data 2 for information on coverage of the individual strains.

We constructed a pseudo-diploid genome by mixing two yeast strains (N44, CBS432) of ANI 98.4%, which are derived from Yeast Population Reference Panel (see Supplementary Data 3). The corresponding real PacBio CLR reads were downloaded from European Nucleotide Archive (ENA) under project PRJEB7245, and we subsampled long reads to match a sequencing coverage of 30× per strain for further analyses. This dataset we refer to as "Yeast pseudo-diploid genome".

We downloaded two real metagenomic datasets (PacBio CLR reads) derived from natural whey starter cultures (NWCs)[42] and mixed both together (see Supplementary Data 3), and then subsampled 20% reads such that we obtained a low-complexity metagenomic dataset, which contains 3 species (6 strains). This dataset we refer to "NWC metagenome".

We downloaded raw long-read sequencing data and the corresponding reference genomes from a 10-plex multiplexed dataset which was sequenced by PacBio Sequel System, Chemistry v3.0. Then, we randomly subsampled 10% reads such that we obtained a mock metagenomic dataset with an average sequencing coverage about 40×, which contains 7 species (9 strains in all, ANI < 98.5%, see Supplementary Data 3). This dataset we refer to "Microbial 10-plex metagenome".

We downloaded the real whole genome sequencing data (Nanopore PromethION) of the human individual HG002 (diploid genome). Subsequently, we performed quality control using fastp[43] and randomly subsampled 50% of the reads for further error correction. The average sequencing coverage per haplotype is about 8x.

We downloaded Nanopore reads of a real human gut microbiome sample from SRA database (accession number: SRR8427258) that was presented in the previous study[44]. The corresponding Illumina reads of the same sample (accession number: SRR6807561, SRR6788327) were also provided in another study[45]. Therefore, we could use Illumina reads to evaluate correction performance without knowing the reference genomes (see below).

## Step 1: Read overlap calculation

Step 1 refers to computation of all-vs-all overlaps for the input reads (first cycle: raw reads, second cycle: pre-corrected reads) using the (widely popular) minimizer based, long-read overlap computation tool Minimap2[29]. During the first cycle, only a seed-chain procedure is performed, while during the second cycle a base-level alignment is added. Because this is very fast, it can manage the large amount of read pairs we need to process easily. Subsequently, bad overlaps are filtered by reasonable, additional criteria, which includes removing overlaps that do not exceed 500 bp in length, self-overlaps, or internal matches. In this, we follow Algorithm 5 in ref. 46 and its implementation in[47]. Additionally, overlaps that have a high error rate, that is $|1 - \min(L_q, L_t) / \max(L_q, L_t)| \geq e$, where $L_q$ and $L_t$ are the lengths of the mapping in the query and the target reads, respectively, and $e$ is the maximum error-rate threshold, are also filtered out[15].

While during the second cycle, the similar procedures (overlap computation and filtration) are also performed but for pre-corrected reads. Unlike in the first cycle, we compute pre-corrected read overlaps with base-level alignment such that the sequence identity of overlaps (overlap identity) can be determined, and then filter overlaps with one more criterion, minimum overlap identity (denoted as $\delta$, $\delta = 0.99$ for simulated sequencing data and $\delta = 0.98$ for real sequencing data). Notably, because most of sequencing errors have been corrected in the first cycle, the sequence identity distribution referring to read overlaps with reads from the same haplotype clearly differs from the sequence identity distribution on read overlaps where the two reads stem from different haplotypes. Therefore, it is

straightforward to filter overlaps that refer to reads from different haplotypes: we filter out read overlaps of sequence identity $< \delta$, where the threshold $\delta$ is a parameter determined based on observations for the error rate of corrected reads after the first cycle. For example, we see that the error rate of corrected reads after the first cycle is basically less than 0.5% for simulated data. Therefore, we could say the error rate of read overlaps (assume two reads are from the same haplotype) is less than $0.5\% \times 2 = 1\%$, that is the overlap identity $\geq \delta = 1 - 1\% = 0.99$.

## Step 2: Read alignment pile generation

Our workflow then selects a target read $r$ and, based on the overlaps computed in step 1, collects reads that overlap $r$. The target read $r$ serves as a backbone, and for each overlap between read $r$ and another read, a fast edit-distance-based alignment[48,49] is then performed, which generates a read alignment pile. The edit-distance-based alignment is only needed to split the read alignment pile into small windows in step 3. Dangling ends of reads that overlap the target read $r$, indicated by horizontal dotted lines in the original read alignment pile in Fig. 1, are removed from further consideration in the following. See[15] for more details.

## Step 3: Window segmentation

The read alignment pile is then divided into several small non-overlapping windows of identical length, with the target read serving as a reference: each such window covers 500 bp of the target read. Obviously, segmenting the read alignment pile reflects a straightforward procedure, because of the pairwise alignments of the target read with its overlapping reads served as the basis for read alignment pile construction, see Step 2[15]. The part of the target read $r$ corresponding with one particular window is further referred to as a 'target subread'. This implies in particular that target subreads are 500 bp in length, apart from the rightmost window, where target subreads can be shorter.

The reason for segmenting the alignment piles into windows of small length is the great reduction in terms of computational burden in the following: the next step 4, as the technical core of our approach being concerned with variation graphs, greatly profits from this segmentation, both in terms of downsizing the original problem as well as in terms of enabling parallelization.

## Step 4: Error correction for target subreads

Step 4 reflects the methodical novelty of VeChat. Step 4 differs when comparing the first with the second cycle, see Fig. 2 for the first and Fig. 3 for the second cycle. Step 4 of the first cycle is considerably more involved, because it reflects the crucial statistical considerations through which to identify sequencing errors. The core idea that underlies these crucial statistical considerations is that true variants are significantly likely to co-occur across different reads, whereas occurrence of errors is random. Correspondingly, the following arguments make sense.

Thanks to dividing reads into subreads, the corresponding computations, such as variation graph construction and statistical evaluation of edges relative to read coverage, can be parallelized across subreads, which speeds up computations substantially.

**Variation graph construction.** The subsequences in a window we refer to as subreads, are subsequently used to construct a variation graph $G = (V, E, P)$. This variation graph is a directed acyclic graph (DAG), where vertices $v \in V$ represent nucleotides (A, T, C, G), edges $(v_i, v_j) \in E$ indicate that the nucleotides represented by nodes $v_i$ and $v_j$ have appeared as a two-letter subsequence in one of the reads from which the graph was constructed, relative to that particular position with respect to the read alignment pile. So, for example, if $v_i$ and $v_j$ correspond to $A$ and $G$, respectively, exactly the reads that relative to the coordinates implied by the target read show $AG$ at that particular

position induce an edge $(v_i, v_j)$. Correspondingly, reads can be identified as certain paths $P = (v_1, ..., v_l)$ in the variation graph. For variation graph construction, we use the partial order alignment (POA) algorithm[30] and its faster version, enhanced by SIMD vectorization, as described in[15].

**Pruning: Principle.** In the following, we will use the notation $v \in V$ to also indicate the letter from the alphabet $\{A, C, G, T\}$ a particular node $v$ refers to, recalling that each node corresponds to exactly one nucleotide in the variation graphs we work with in the following.

The high error rate affecting TGS reads and the possible bias introduced by constructing the graph (because, for example, the order relative to which reads are considered has an influence), the variation graph constructed in the first cycle contains many spurious vertices and edges. For pruning the graph from mistaken edges and/or nodes (vertices), we adopt techniques from frequent itemset mining. The basic idea is to identify edges with itemsets, and to prune edges from the graph if the corresponding itemsets do not appear to be sufficiently frequent. After removal of edges, reads are realigned against the resulting graph, such that itemset counts have to be re-computed. This may render more edges to correspond to itemsets that are not sufficiently frequent. The cycle of identifying edges as infrequent itemsets, removing them, and re-aligning reads is repeated until convergence, that is, until no further edges are identified as spurious. In practice, we determined 3 as an appropriate number of iterations for our experiments. Note that for re-aligning reads against the modified graph, we make use of the POA algorithm, without, however, re-modifying the graph anymore.

The model that underlies the mining of frequent itemsets is the "market-basket model". Baskets correspond to sets of items, and frequent itemsets correspond to subsets of items that appear in sufficiently many baskets, or, vice versa, infrequent itemsets correspond to subsets of items that do not appear in sufficiently many baskets.

Following this model, the basic set of items agrees with the set of nodes $V$ in the variation graph. Baskets then correspond to reads, which are modeled as paths $P = (v_1, ..., v_l)$, and the items they contain correspond to the nodes $v_1, ..., v_l \in V$ the reads cover. Further, the itemsets we are interested in correspond to edges $e = (v, w)$, as pairs of items $v, w$. If two nodes $v, w$ that are connected by an edge $e = (v, w)$, as particular subsets of items, do not appear in sufficiently many baskets, that is, the corresponding edge $e = (v, w)$ is not contained in sufficiently many read induced paths $P$, the corresponding edge is removed from the graph.

For appropriately quantifying "sufficiently many baskets", we make further use of Support and Confidence as two standard definitions from frequent itemset mining. "Support" just corresponds to the number of baskets a particular subset of items is contained in. Here, Support just agrees with the number of reads by which a particular edge $e = (v, w)$ is covered. "Confidence", on the other hand, corresponds to measuring whether appearance of items in a basket is correlated with other items appearing in that basket. Here, Confidence corresponds to the amount of reads that cover the edge $(v, w)$ in relation to how many reads cover $v$, on the one hand, and in relation to how many reads cover $w$, on the other hand. If neither sufficiently many reads that cover $v$ also cover $(v, w)$, nor sufficiently many reads that cover $w$ also cover $(v, w)$, we "loose confidence" in $e = (v, w)$, because the edge $(v, w)$ could reflect sequencing error noise.

**Pruning: definitions.** To make the ideas from above explicit, let $R(v)$ be all reads that cover node $v$. For $r \in R(v)$, let further $p_{r,v}$ reflect the probability that $v$ reflects an error in $r$. When dealing with FASTQ files, the probability $p_{r,v}$ is derived from the Phred profile of $r$. In case of FASTA files, $p_{r,v}$ is taken as zero.

We now would like to determine $w(v)$, as a weight for node $v$ that reflects the expected number of reads that cover it. Note that for

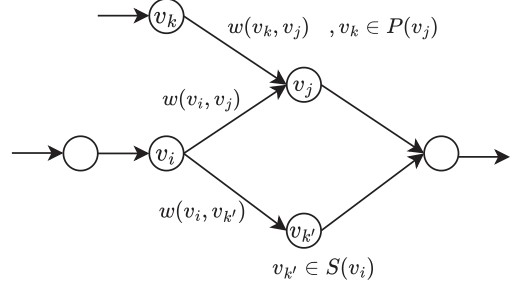

**Fig. 4 | A schematic diagram for explaining the calculation of Support and Confidence for edge $(v_i, v_j)$.** $w(v_i, v_j)$ is the approximation for the expected amount of reads that cover edge $(v_i, v_j)$. $S(v_i)$ and $P(v_j)$ denote the vertices that succeed $v_i$ and precede $v_j$, respectively. In this directed graph, $v_j, v_{k'}$ are the successors of $v_i$, whereas $v_i, v_k$ are the predecessors of $v_i$.

FASTA files, $w(v)$ just agrees with the number of reads that cover $v$. For FASTQ files, $w(v)$ corresponds to summing up $1 - p_{r,v}$ across all reads $r \in R(v)$, as the sum of the probabilities that the reads $r \in R(v)$ indeed reflect the letter associated with $v$. In terms of formulas, we obtain

$$w(v) : = \begin{cases} \sum_{r \in R(v)} 1 & \text{for FASTA files} \\ \sum_{r \in R(v)} 1 - p_{r,v} & \text{for FASTQ files} \end{cases} \quad (1)$$

For an edge $e = (v_i, v_j)$, let $R(v_i, v_j)$ be all reads that cover edge $e$. We further determine

$$w(e) = w(v_i, v_j) : = \frac{1}{2} \sum_{r \in R(v_i, v_j)} w(v_i) + w(v_j) \quad (2)$$

as an approximation for the expected amount of reads that cover $e = (v_i, v_j)$. Note that $w(e)$ corresponds to exactly the amount of reads that cover $e$ for FASTA files. For FASTQ files, the expected amount of reads that cover $(v_i, v_j)$ virtually corresponds to the sum of products $(1 - p_{r,v_i})(1 - p_{r,v_j}) = 1 - p_{r,v_i} - p_{r,v_j} + p_{r,v_i} p_{r,v_j}$ (*) across $r \in R(v_i) \cap R(v_j)$. Here, for speeding up computations, we opt for approximating (*) by $1 - \frac{1}{2} p_{r,v_i} - \frac{1}{2} p_{r,v_j}$, which reflects (2). Since this agrees with (*) in terms of orders of magnitude, this introduces only negligible deviations from the true values; experiments of ours confirmed that the gain in speed offset the loss in precision on that account.

Based on these weights, we now define the two metrics Support and Confidence. See also Fig. 4 for an illustration of the relevant definitions. In formal detail, let $e = (v_i, v_j)$ be an edge. Let then Support of $e$ be defined as

$$\text{Support}(e) : = w(e) \quad (3)$$

that is, just as the weight of $e$. Further, note that Confidence is an asymmetrical measure: the probability to observe $v_j$ in a read that contains $v_i$ may disagree with the probability to observe $v_i$ in a read that contains $v_j$. We take this into account by defining

$$\text{Confidence}(v_i, v_j) : = \text{Confidence}(v_i \rightarrow \{v_i, v_j\}) : = \frac{w(v_i, v_j)}{\sum_{v_{k'} \in S(v_i)} w(v_i, v_{k'})} \quad (4)$$

on the one hand, and

$$\text{Confidence}(v_j, v_i) : = \text{Confidence}(v_j \rightarrow \{v_i, v_j\}) : = \frac{w(v_i, v_j)}{\sum_{v_k \in P(v_j)} w(v_k, v_j)} \quad (5)$$

on the other hand, where $S(v_i)$ and $P(v_j)$ denote the vertices that succeed $v_i$ and precede $v_j$, respectively (and where $v_i \rightarrow \{v_j, v_j\}$ and $v_j \rightarrow \{$

$v_i, v_j$} agree with standard notation from association rule mining). We eventually declare

$$\text{Confidence}(e) := \max\{\text{Confidence}(v_i, v_j), \text{Confidence}(v_j, v_i)\} \quad (6)$$

as the overall Confidence in $e = (v_i, v_j)$.

It remains to determine appropriate thresholds $s$ and $c$, such that edges $e$ for which either Support$(e) < s$ or Confidence$(e) < c$ are pruned from the graph. Note that by its definition, Confidence reflects the probability that a read that covers $v_i$ also covers $v_j$, or vice versa. We determined $c = 0.2$ as an appropriate threshold in experiments; see the Supplementary Figure 1 for the corresponding outcome.

Support, however, does not reflect a probability. Depending on the overall amount of reads in a subwindow, and the length of a sub-window—that is virtually depending on the average read coverage of a subwindow—the Support needs to be appropriately scaled. Therefore, consider that $C := \sum_{v \in V} w(v)/L$ is the average coverage of a position in the subwindow. Accordingly, we determine $s := 0.2 \times C$ as a threshold that takes subwindow specific coverage into appropriate account.

Note eventually that both Support and Confidence are required for effective pruning of the graph, see Supplementary Figure 2 for the correlation between the two quantities.

**Optimal alignment path extraction.** Upon convergence of the pruning algorithm, the target subread that corresponds to a small window is realigned against the fully pruned variation graph that results from the last iteration of the pruning algorithm. The path in the graph that corresponds to the optimal alignment of the target subread is then taken as the pre-corrected target subread; see the orange elements in Fig. 2 for an illustration.

### Step 5: Concatenation
In this step, pre-corrected target subreads are concatenated to a whole, pre-corrected target read, which corresponds to the obvious, straightforward idea of "patching together" pre-corrected target sub-reads; see "5. Concatenation" in Fig. 1 for an illustration.

### Step 6: Merging corrected target reads
Steps 2–5 are repeated until all reads have been corrected. Step 6 then reflects a simple operation: the overall set of pre-corrected reads that result from the repeated execution of steps 2–5 steps are merged and taken as input for the second cycle.

### Second cycle: Modifications Step 1 and 4
Note that the pre-corrected reads generated by way of cycle 1 still contain a small, but yet non-negligible amount of random errors (see Supplementary Table 14). Cycle 2 addresses to correct these remaining errors. To do so, steps 1–6 are repeated with, however, some crucial modifications in steps 1 and 4. See the blue elements in Fig. 1 for the workflow that reflects the procedures of cycle 2. To be specific, the modification of step 1 consists in not only computing all-vs-all overlaps based on minimizers, but also base-level alignments[29] for the pre-corrected reads. This considerably facilitates to filter reads overlaps according to which they stem from identical haplotypes, based on sequence identity related thresholds. We use $\delta = 0.99$ for sequencing error rates of 5–10% and $\delta = 0.98$ for a sequencing error rate of 15% in our experiments, which we also generally recommend.

The modification of step 4 then relates to generating a single sequence from each variation graph, instead of performing iterative graph pruning and sequence-to-graph re-alignment. For that, the dynamic programming algorithm referred to as "heaviest bundle algorithm" in ref. 31, as indicated in Fig. 3 is used. Note that generating a single sequence from each of the local variation graphs is reasonable in the second cycle, because the overlapping reads used to correct the target read can already be assumed to stem from the same haplotype.

Finally, we obtain fully error-corrected reads as the output of the second cycle; see "All final corrected reads (cycle2)" in Fig. 1.

### Benchmarking: alternative approaches
To enable a fair and meaningful comparison, we considered all popular state-of-the-art tools that perform TGS read self-correction. Namely, this selection includes Racon (v1.4.13)[15], CONSENT (v2.2.2)[20], Canu (v2.1.1)[16] and Daccord (v0.0.18)[18]. We ran all tools using their default parameters (command details are provided in the Supplementary Methods).

Apart from just evaluating error correction, we were also interested in the effects when using the corrected reads for genome assembly, as the prior, canonical area of application for corrected reads. Therefore, we considered Canu (v2.1.1) and (meta)Flye (v2.8.2), which can perform haplotype-aware assembly or metagenome assembly, as well-known assemblers, with or without prior error correction by VeChat. Also, because the corrected reads have error rates of less than 1%, one can run HiCanu (Canu in HiFi mode, which applies for the given error rates) on the reads, which we included into our considerations.

### Metrics for evaluation
We evaluated genome assembly performance by means of several commonly used metrics, as reported by QUAST V5.1.0[50]. See below for specific explanations, and see http://quast.sourceforge.net/docs/manual.html for full details. Note that the principled qualities of error-corrected long reads are covered by standard QUAST criteria as well; for example, low haplotype coverage reflects that true variants were mistakenly identified as errors, while error rates and mis-assemblies reflect that the correction procedure overlooked errors, or even confounded reads in terms of their origin through mistaken correction. As usual, corrected reads and contigs of length less than 500 bp were filtered from the output before evaluation. Note that we ran QUAST with the option `-ambiguity-usage one`, which appropriately takes into account that our datasets reflect mixed samples (such as polyploid genomes or metagenomes). In addition, for evaluating the real human genome (HG002) and human gut microbiome data, there is no ground truth of the reference genomes. Hence, we applied Merqury[32] to evaluate results based on auxiliary Illumina sequencing reads, in a reference-free manner.

Error rate (ER). The error rate is equal to the sum of mismatch rate and indel rate when mapping the obtained corrected reads or contigs to the reference haplotype sequences. When reference genomes are unknown, consensus quality value (QV) and switch error rate reported from Merqury are used as alternative metrics of approved usefulness[21,51,52]. Note that the QV metric mainly reflects the base quality but does not consider long-range switch errors because it relies on short k-mer strategies (k = 21 by default).

Haplotype coverage (HC). Haplotype coverage is the percentage of aligned bases in the ground truth haplotypes covered by corrected reads or contigs, which is used to measure the completeness of the assembled contigs or the corrected reads. The k-mer completeness from Merqury is also used as an alternative metric when performing reference-free evaluation.

N50 and NGA50. N50 is defined as the length for which the collection of all corrected reads/contigs of that length or longer covers at least half the given sequences. NGA50 is similar to N50 but can only be calculated when the reference genome is provided. NGA50 only considers the aligned blocks (after breaking reads/contigs at misassembly events and trimming all unaligned nucleotides), which is defined as the length for which the overall size of all aligned blocks of this length or longer equals at least half of the reference haplotypes. Both N50 and NGA50 are used to assess the length distribution of corrected reads and the contiguity of the assemblies. Note that this may be of relatively little interest for corrected reads. We nevertheless display

corresponding results because error correction does have an influence on these statistics.

Number of misassemblies (#Misassemblies). The misassembly event in corrected reads or assemblies indicates that left and right flanking sequences align to the true haplotypes with a gap or overlap of more than 1kbp, or align to different strands, or even align to different haplotypes or strains. Here, we report the total number of mis-assemblies in the given sequence data.

## Reporting summary

Further information on research design is available in the Nature Portfolio Reporting Summary linked to this article.

## Data availability

The authors declare that all of the datasets used in this paper are publicly available. The simulated and real (including Yeast pseudo-diploid genome and NWC metagenome data) long-read raw sequencing data used for benchmarking experiments are publicly available in Zenodo under 10.5281/zenodo.5501454 (https://doi.org/10.5281/zenodo.5501455)[53]. The real sequencing data of Microbial 10-plex metagenome was downloaded from https://downloads.pacbcloud.com/public/dataset/microbial_multiplex_dataset_release_SMRT_Link_v6.0.0_with_Express_2.0/. The real whole genome sequencing data of human individual HG002 (diploid genome) was downloaded from https://s3-us-west-2.amazonaws.com/human-pangenomics/NHGRI_UCSC_panel/HG002/hpp_HG002_NA24385_son_v1/nanopore/downsampled/standard_unsheared/HG002_ucsc_Jan_2019_Guppy_3.4.4.fastq.gz. The Nanopore reads[44] of the real human gut microbiome sample were downloaded from SRA database under accession number: SRR8427258 (https://www.ncbi.nlm.nih.gov/sra/?term=SRR8427258), and the corresponding Illumina reads of the same sample[45] were from accession numbers (https://www.ncbi.nlm.nih.gov/sra/SRX3765823): SRR6807561, SRR6788327. Genome descriptions of simulated sequencing datasets are shown in Supplementary Data 1. Sequencing coverage information is shown in Supplementary Data 2. Genome descriptions of real sequencing datasets are shown in Supplementary Data 3. Raw data for drawing Supplementary Fig. 1 and 2 are provided with this paper and are available in Zenodo under 10.5281/zenodo.7239028 (https://doi.org/10.5281/zenodo.7239028).

## Code availability

The source code of VeChat is GPL-3.0 licensed, and publicly available at https://github.com/HaploKit/vechat. The results presented in this study can be reproduced from Code Ocean under DOI: 10.24433/CO.2329278.v2 (https://codeocean.com/capsule/7010505/tree/v2)[54].

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

## Acknowledgements

X.L. and X.K. were supported by the Chinese Scholarship Council. A.S. was supported by the Dutch Scientific Organization, through Vidi grant 639.072.309 during the early stages of the project, and from the European Union's Horizon 2020 research and innovation program under Marie Skłodowska-Curie grant agreements No 956229 (ALPACA) and No 872539 (PANGAIA).

## Author contributions

X.L. and A.S. developed the method. X.L. implemented the software and conducted the data analysis. X.K. simulated the metagenomic datasets. X.L. and A.S. wrote the manuscript. All authors read and approved the final version of the manuscript.

## Funding

## Competing interests

The authors declare no competing interests.
