## [Peer Review File · Nature Communications]

VeChat: Correcting errors in long reads using variation graphsREVIEWER COMMENTS

Reviewer #1 (Remarks to the Author):

In “VeChat: Correcting errors in long reads using variation graphs” authors Luo, Kang, and Schönhuth provide a variation graph based read error correction system. This method progressively refines a variation graph model of small windows of the all-to-all alignment of the read set, and then corrects read alignments by mapping to these. The authors provide extensive benchmarking based on reasonable simulated and synthetic data sets. These results make a strong case that VeChat is capable of achieving the authors state goal of allowing read correction while accounting for variation in the underlying sample caused by heterozygosity, metagenomic, or pangenomic contexts.

I enjoyed the manuscript and found it to be well written. I have a handful of suggestions for presentation, a request for improved experimental studies, and some experience that suggests potential limitations of the method that perhaps the authors could address.

In terms of presentation, I felt the initial description of exactly how a read was corrected to be somewhat high-level and difficult to follow. In the methods, the description becomes clear. I suggest highlighting that the path of the optimal alignment through the VG is taken to be the corrected read earlier in the text. Additionally, the process of pruning is initially presented somewhat vaguely. It is always hard to strike a good balance with these things, but perhaps some adjustment of the high level descriptions could be made to improve the reader’s intuition about what’s going on.

The experimental setup is excellent, and I think this kind of foundation is required to demonstrate the method. Especially the issue of handling high ploidy seems to be difficult to tackle otherwise. I did wonder if there is a possible approach that works on a non-synthetic data set. For instance, could the authors correct reads from HG002 and then measure the read accuracy via alignment to the reference and concordance with the genome in a bottle NIST truth set for the sample?

We also attempted to apply VeChat. We have been applying GraphAligner to correct ONT reads from a population of ~50k phage genomes to a combined de Bruijn graph built from all (even short) ONT reads. In this context, we didn’t get very good results with VeChat. I wonder if the dependence on POA causes issues with extremely high depths, and if the authors could attempt to replicate (in a simulated context) something like what we are approaching. What happens when the data set has a “ploidy” of ~1000X, over a small genome? We can certainly share our data if the authors are interested, however it does lack ground truth and I think this might just be an extreme case that is beyond the testing that the authors have demonstrated.

Should the authors reference other works in the variation graph space (including their own), if only for the purpose of linking their work into a longer arc of research? This is not critical! In my opinion, not citing indicates that the authors believe that the variation graph concept is simple and generic enough to stand on its own.

Erik Garrison

Reviewer #2 (Remarks to the Author):

Overview:

=====

Correction of sequencing errors is a common first step in analysing PacBio or ONT data, especially when the target application is genome assembly. The authors present a new method for error correction, VeChat. VeChat first finds overlaps between the reads. It then identifies sets of overlapping reads and constructs a variation graph from them. Finally the reads are corrected by aligning them to the graph. VeChat proceeds in two iterations with the same overall structure but subtle differences in the details.

The idea of building a graph based representation of reads and then aligning reads to it to correct them has been presented before for hybrid error correction (see LoRDEC, Jabba and GraphAligner for examples). These approaches use DBGs instead of variation graphs to represent the reads but the DBG will include the variation present in the reads. However, I am not aware that this approach would have been applied to self-correction successfully and the hybrid methods were not particularly designed for preserving variation.

The novelty of this paper is to devise a method for building a variation graph based on the overlaps of the reads and to show that the approach works in self-correction and preserves variation.

The paper is well written. I was able to install the software and run a small example provided in the repository.

Suggestions for revision:

=====

Major

1) The authors claim in the abstract and introduction that the current error correction methods all are based on raising a consensus sequence which reflects the summary of the observed reads. I find this a bit misleading as several hybrid methods such as LoRDEC, Jabba, and GraphAligner in fact represent the short reads with a de Bruijn graph and correct the long reads by aligning them to the de Bruijn graph. It is true that these methods were not designed specifically for preserving variation or to preserve low-frequency haplotypes/strains but still the approach is quite similar to the one presented in this paper.

2) Metagenome data, lines 198-205: I think the most crucial parameter for the success of error correction is the sequencing coverage of the lowest abundance strain. Thus I would like to see here what is the range of sequencing coverage over the different strains, i.e. what is the sequencing coverage of the minimum abundance strain / the maximum abundance strain?

3) The definition of the Support of an edge is not clear to me. On lines 512-513, it is stated that Support of an edge agrees with the number of reads covering that edge. However, on lines 506-509 it is not clear to me if it is enough that the nodes v and w appear in that order in a read induced path or if the nodes v and w need to be adjacent in that path, i.e. the edge (v,w) belongs to the path. The definition of the weight of an edge (Equation 2) seems to indicate that the nodes v and w do not even need to appear in that order in a read induced path but it would be enough for them to belong to the path. If the adjacency of the nodes is not required, this would imply that erroneous edges inserted to the graph due to deletion errors in the reads will have high weights because most read induced paths will visit both end points of the edge although they do not include the edge itself.

4) The authors write that the pre-corrected reads generated by way of cycle 1 still contain a small, but yet non-negligible amount of random errors. It would be interesting to see the benchmarking results for some data set for the pre-corrected reads. This would help to evaluate what is the significance of cycle 2 in the method.

Minor

5) Page 11, line 322: An incomplete sentence: "Here, we report on quick application"

6) Page 17, line 570: "set of pre-corrected" -> "set of pre-corrected reads"

Optional

7) Section "Workflow", pages 3-5: Mentioning that the variation graph is constructed with the POA algorithm already in this section would help readers who are familiar with the methods in this area to quickly grasp the ideas. In the current manuscript, this of course becomes clear in the Methods section.

REVIEWER COMMENTS

Reviewer #1 (Remarks to the Author):

In “VeChat: Correcting errors in long reads using variation graphs” authors Luo, Kang, and Schönhuth provide a variation graph based read error correction system. This method progressively refines a variation graph model of small windows of the all-to-all alignment of the read set, and then corrects read alignments by mapping to these. The authors provide extensive benchmarking based on reasonable simulated and synthetic data sets. These results make a strong case that VeChat is capable of achieving the authors state goal of allowing read correction while accounting for variation in the underlying sample caused by heterozygosity, metagenomic, or pangenomic contexts.

I enjoyed the manuscript and found it to be well written. I have a handful of suggestions for presentation, a request for improved experimental studies, and some experience that suggests potential limitations of the method that perhaps the authors could address.

Comment: In terms of presentation, I felt the initial description of exactly how a read was corrected to be somewhat high-level and difficult to follow. In the methods, the description becomes clear. I suggest highlighting that the path of the optimal alignment through the VG is taken to be the corrected read earlier in the text. Additionally, the process of pruning is initially presented somewhat vaguely. It is always hard to strike a good balance with these things, but perhaps some adjustment of the high level descriptions could be made to improve the reader’s intuition about what’s going on.

Response: To improve the understandability of the higher level descriptions, we have added one paragraph right at the beginning of the subsection ‘Workflow’:

“The basic idea of VeChat is to construct a variation graph from the all-to-all alignments of the raw reads. One then identifies nodes and edges in the resulting graph that are likely to be artifacts, and removes them. Subsequently, reads are re-aligned against the resulting, pruned graph. The path in the pruned graph corresponding to the optimal re-alignment points out an error-corrected sequence of the read. The procedure of spurious node and edge removal followed by re-alignment is repeated until convergence (note that the statistical evaluation of remaining nodes and edges changes upon re-alignment, which may reveal new likely spurious nodes and edges in the next iteration). The re-alignment of the original read with the final graph points out the fully error corrected sequence of the read.”

We have also added an extra sentence in the fourth paragraph of the subsection ‘Workflow’:
“The path in the pruned variation graph that corresponds to the optimal alignment of the target subread is then taken as the pre-corrected target subread;”

Comment: The experimental setup is excellent, and I think this kind of foundation is required to demonstrate the method. Especially the issue of handling high ploidy seems to be difficult to tackle otherwise. I did wonder if there is a possible approach that works on a non-synthetic data set. For instance, could the authors correct reads from HG002 and then measure the read accuracy via alignment to the reference and concordance with the genome in a bottle NIST truth set for the sample?

Response: We thank you for pointing out the experimental setup that enables us to evaluate our method on non-synthetic data. We have followed your instructions and have run VeChat on both HG002 (diploid) and a real human gut microbiome dataset (see “Datasets” section in the manuscript). Reads from both data sets were sequenced using Oxford Nanopore. Results have been collected into the additional Supplementray Table S2, see further below also here.

As for evaluating the real human genome (HG002) and human gut microbiome data, there is no ground truth of the reference genomes. Hence, we applied Merqury (Rhie, A., Walenz, B.P., Koren, S. et al. 2020) to evaluate results based on auxiliary Illumina sequencing reads, in a reference-free manner.

In a first (more generally important?) remark note that Merqury is affected by substantial biases with respect to the method whose results are evaluated. The effects persist across datasets and sequencing technologies, see the following table (Supplementray Table S3 in the manuscript). This makes us assume that the biases are indeed due to the underlying error correction methodology:

Method	Metrics from Merqury			Metrics from QUASt	
	Error rate (% , QV)	Switch error (%)	k-mer completeness (%)	Error rate (%)	Haplotype coverage (%)
E.coli genomes (Ploidy=2,PacBio)					
VeChat	0.008	0.16	100.0	0.014	100.0
CONSENT	0.041	6.88	99.9	0.194	99.9
Racon	0.102	5.45	98.5	0.276	99.2
Canu	0.137	4.92	99.7	0.308	99.9
Daccord	0.022	5.46	92.4	0.423	99.2
E.coli genomes (Ploidy=2,ONT)					
VeChat	0.014	0.16	99.9	0.022	99.9
CONSENT	0.049	6.95	99.9	0.212	99.9
Racon	0.136	6.17	98.1	0.346	99.3
Canu	0.190	5.44	99.7	0.390	100.0
Daccord	0.037	6.07	92.7	0.438	99.2
Metagenome (Low complexity,PacBio)					
VeChat	0.015	-	98.4	0.036	96.9
Racon	0.082	-	95.8	0.200	91.7
CONSENT	0.055	-	98.4	0.214	98.4
Canu	0.122	-	98.4	0.259	97.4
Daccord	0.013	-	90.5	0.259	92.8
Metagenome (High complexity,PacBio)					
VeChat	0.029	-	96.4	0.088	97.5
CONSENT	0.096	-	98.8	0.274	99.4
Canu	0.163	-	98.6	0.354	99.0
Racon	-	-	-	-	-
Daccord	-	-	-	-	-

Table S3. Comparison between reference-free (Merqury) and reference-based (QUAST) evaluations. We performed error correction benchmarking experiments on 4 simulated datasets (pseudo-diploid *E.coli* genome and metagenome datasets as we described in the 'Datasets' subsection), of which the ground truth are known. QV: consensus quality value. Methods are sorted by the values of column 'Error rate' (QUAST).

A closer look reveals that Merqury quite evidently favors CONSENT and Daccord over Racon, Canu and VeChat (where VeChat appears to be the least favored method). Since both CONSENT and Daccord are de Bruijn graph based, and Merqury bases evaluation statistics

on k-mer related counts, an immediate hypothesis is that Merqury is biased towards evaluating de Bruijn graph based tools more favorably. At any rate, the results achieved using Merqury and displayed in Supplementary Table S2 (see below) require to be discussed with considerable caution.

Method	#Reads	Error rate (% , QV)	Switch error (%)	Haplotype coverage (%)	N50 (bp)
HG002 (diploid)					
VeChat	2385346	0.328	11.6	92.2	53620
CONSENT	2578729	0.478	13.6	97.0	54298
Canu	-	-	-	-	-
Daccord	-	-	-	-	-
Racon	-	-	-	-	-
Human gut microbiome					
Daccord	3050430	1.323	-	91.8	4073
VeChat	2733544	1.352	-	91.7	3622
CONSENT	3097015	1.958	-	93.9	4672
Canu	1705153	2.351	-	93.3	5399
Racon	-	-	-	-	-

Table S2. Error correction benchmarking results for real ONT sequencing data (non-synthetic). For the human genome sample HG002, we stopped Canu after running 22 days on a computer with 160 CPUs and 2TB RAM due to out of disk space (the size of temporary files > 10TB), so no result was reported. Daccord and Racon failed to run for HG002 dataset. Racon failed to run for the human gut microbiome data. Note that Merqury is used to evaluate the error correction performance of these two datasets since the ground truth is unknown. The ‘error rate’ and ‘haplotype coverage’ correspond to the ‘consensus quality value (QV)’ and ‘k-mer completeness’ reported by Merqury, respectively. The ‘switch error’ of human gut microbiome data is unavailable.

Evidently, VeChat’s advantages are less obvious than on synthetic data. Nevertheless, still VeChat achieves the best error rates on HG002, at the expense of reduced haplotype coverage, and is outperformed by miniscule margins on microbiome data by Daccord, which however is unable to correct errors in HG002 data.

We have added some paragraphs in the Results section:

“Real sequencing data (Non-synthetic, ONT). See Supplementary Table S2 for error correction benchmarking experiments for real ONT sequencing data, which have been evaluated using Merqury (Rhie, A., et al. 2020) because of the lack of reference genomes. Before discussing results, see Supplementary Table S3, which puts evaluations with and without a reference genome (QUAST resp. ~Merqury), that is, with and without available ground truth into context. Corresponding results immediately point out that Merqury is subject to substantial biases with respect to the choice of methods. For example, on ONT sequenced diploid genomes, Merqury underestimates the true error rates (as performed by QUAST relative to the ground truth) by factors of 4.33 (CONSENT) and even 11.84(!) (Daccord), but only by factors of 2.54 (Racon), 2.05 (Canu), and 1.75 (VeChat).

The quality of the results persists on the other datasets: Merqury evidently favors CONSENT and Daccord quite substantially in comparison with Racon, Canu and VeChat. Because Merqury is k-mer based, an immediate hypothesis is that Merqury tends to favor k-mer (e.g. ~de Bruijn graph) based approaches (CONSENT, Daccord) over approaches that do not make use of de Bruijn graphs (Racon, Canu and VeChat), where VeChat appears to be the only tool whose error rates are not substantially underestimated, at least on the lesser complex datasets. In summary,

we have experienced that Merqury is affected by considerable volatility with respect to the methodological background of error correction tools, clearly favoring certain tools over others.

Therefore, the discussion of the following results are to be taken with the corresponding caution in terms of the method specific biases that Merqury appears to induce.

As becomes obvious from Supplementary Table S2 VeChat achieves approximately 1.5 times lower error rate (QV) and 1.2 times lower switch error on HG002 compared with CONSENT (the only alternative tool available to compare), while losing more haplotype coverage. Whereas on the human gut microbiome data set, Daccord achieves the lowest error rate (QV) while VeChat obtains comparable read accuracy. VeChat achieves about 1.4 and 1.7 times lower error rates (QV) in comparison to CONSENT and Canu, respectively, while keeping comparable performance in terms of other aspects. (We recall that Daccord was the tool whose error rate was underestimated by the by far largest factors, which points out that, potentially, VeChat virtually achieves better error rates).

As we mentioned earlier in the subsection 'Metrics for evaluation', the error rate (QV) ignores long-range switch errors in evaluation, which is unable to represent the overall error rate of reads. In fact, in simulated datasets of which the ground truth are known, we observed that VeChat achieves much lower switch error rate compared with others, and in both metagenome datasets VeChat achieves much lower overall error rate (from QUAST), even though its error rate (QV, from Merqury) is comparable with Daccord in the metagenome dataset of low complexity (see Supplementary Table S3). In summary, we speculate VeChat can achieve better performance in terms of overall error rate on the real human gut microbiome data."

Comment: We also attempted to apply VeChat. We have been applying GraphAligner to correct ONT reads from a population of ~50k phage genomes to a combined de Bruijn graph built from all (even short) ONT reads. In this context, we didn't get very good results with VeChat. I wonder if the dependence on POA causes issues with extremely high depths, and if the authors could attempt to replicate (in a simulated context) something like what we are approaching. What happens when the data set has a "ploidy" of ~1000X, over a small genome? We can certainly share our data if the authors are interested, however it does lack ground truth and I think this might just be an extreme case that is beyond the testing that the authors have demonstrated.

Response: Thank you for your efforts, and testing VeChat in a "high sequencing coverage over a small genome" scenario.

Coincidentally, we originally tried to develop and run the idea of VeChat on virus quasispecies data with ultra-high sequencing coverages (typically with total sequencing coverage of 20000x, hundreds to thousands of x per strain), and were planning to integrate such an error correction protocol in our previous study "Strainline: full-length de novo viral haplotype reconstruction from noisy long reads" (<https://doi.org/10.1186/s13059-021-02587-6>). The idea of VeChat works in this case, but Daccord achieves the best performance, see the table below (from Luo et al., 2022):

	No. of reads	Haplotype coverage(%)	N50 (bp)	Error rate (%)	Mismatch(%)	Indel(%)
Daccord	51823	99.9	2902	0.023	0.011	0.011
CONSENT	102931	99.8	2315	0.989	0.020	0.969
Canu	96119	100.0	2384	1.167	0.638	0.529
LoRMA	46463	99.9	2777	1.349	0.187	1.162

Table S4. Benchmarking results for error correction tools on simulated 5-strain HIV mixture PacBio CLR data. The total sequencing coverage in this table is 20000×.

Motivated by the idea that Daccord works excellently for data from short genomes (apparently not noticed by the authors themselves), but does not work well for longer genomes (as originally intended by the authors), we developed VeChat as a tool that one can apply for polyploid and mixed sample / metagenome settings not characterized by ultra-high coverage.

When processing ultra-high sequencing coverage data, VeChat runs into the following issues: (1) excessive amounts of read overlaps will consume too much time; (2) as mentioned, VeChat's dependence on POA leads to distorted variation graphs when confronted with too many reads.

For completion, we nevertheless benchmarked VeChat and other methods on a simulated dataset that reflects what you were aiming to do. For that, we simulated a "5-strain HIV mixture" dataset. Please see below for the details about the data description and the benchmarking results:

Method	#Reads	Error rate (%)	Mismatch (%)	Indel (%)	Haplotype coverage (%)	N50 (bp)
Simulated PacBio CLR						
VeChat	26162	0.208	0.132	0.076	99.6	1974
GraphAligner	26159	0.383	0.381	0.002	99.9	2626
Daccord	13020	0.565	0.542	0.023	99.8	2908
CONSENT	25738	0.687	0.022	0.665	99.6	2310
Canu	18038	1.345	0.257	1.088	99.9	2678
Racon	26162	1.487	0.783	0.704	99.1	2194
Simulated ONT						
CONSENT	24762	0.078	0.003	0.075	99.8	2569
VeChat	24957	0.403	0.246	0.157	99.6	1999
GraphAligner	24976	0.442	0.438	0.004	100.0	2817
Daccord	25824	0.527	0.384	0.143	99.9	2163
Canu	18271	1.009	0.262	0.746	100.0	2913
Racon	26018	1.057	0.502	0.555	99.2	2179

Table S5. Benchmarking results of error correction tools on the simulated 5-strain HIV mixture (genome size ≈10Kbp) dataset. This dataset is one of the most challenging datasets in viral quasispecies assembly, which consists of five known HIV-1 strains (YU2, NL43, JRC5F, HXB2, 896) and has been used for benchmarking experiments in many related studies, such as (Giallonardo *et al.*, 2014; Baaijens *et al.*, 2017, 2019; Luo *et al.*, 2022). The overall sequencing coverage of the virus data in this table is 5000×. The sequencing coverage of each virus strain varies from 500× to 1500× (average coverage is about 1000×). Note that GraphAligner performs hybrid error correction (i.e., using both short and long reads) and requires a de Bruijn graph as input. Thus, we first simulated Illumina short reads (2x150 bp) with the same sequencing coverage for each strain using ART (Huang *et al.*, 2012), then we constructed a de Bruijn graph from short reads using BCALM2 (Chikhi *et al.*, 2016) and finally we corrected long read sequencing errors with GraphAligner (Rautiainen and Marschall, 2020). Technically, the results of GraphAligner are incomparable here since all other methods perform self correction.

The results show that VeChat outperforms others on the PacBio data, while it achieves the second best performance on the Nanopore data (slightly better than GraphAligner).

We have added the following paragraph in the subsection "Varying read coverage" of "Results" section:

“In addition, we particularly tested VeChat in the scenario of ultra-high sequencing coverage over a small genome. To reflect this context, we simulated a 5-strain HIV mixture (genome size = 10Kbp) dataset, which has been used for benchmarking experiments in many related studies, such as (Baaijens et al., 2019; Giallonardo et al., 2014; Baaijens et al., 2017; Luo et al., 2022b). The average sequencing coverage per strain is about 1000x. See the Supplementary Table S5 for the details about the data descriptions and the benchmarking results. The results show that VeChat outperforms others on the PacBio data in terms of error rate, whereas it achieves the second best performance on the Nanopore data. In summary, VeChat still works for the extremely high sequencing coverage case, but is not necessarily very effective especially for ONT reads.”

Comment: Should the authors reference other works in the variation graph space (including their own), if only for the purpose of linking their work into a longer arc of research? This is not critical! In my opinion, not citing indicates that the authors believe that the variation graph concept is simple and generic enough to stand on its own.

Response: We do agree with the idea to link our work into a longer arc of research, because we feel like acting in that tradition. We are now citing the paper “Paten, B., Novak, A.M., Eizenga, J.M. & Garrison, E. Genome graphs and the evolution of genome inference. *Genome Res.* 27, 665–676 (2017).” in the Introduction (Line 93), to clarify where the concept of a ‘variation graph’ originates from.

In addition, we are now including the following text in the ‘Introduction’: “*Variation graphs have been effectively used to solve various problems in computational genomics, such as improving the mapping of reads and calling variant (Garrison et al., 2018; Martiniano et al., 2020; Siren et al., 2021), modelling haplotypes (Rosen et al., 2017) and assembling genomes from mixed samples (Baaijens et al., 2019, 2020). To the best of our knowledge, VeChat is the first approach to apply variation graphs for long read error correction.*”

Erik Garrison

Reviewer #2 (Remarks to the Author):

Overview:

=====

Correction of sequencing errors is a common first step in analysing PacBio or ONT data, especially when the target application is genome assembly. The authors present a new method for error correction, VeChat. VeChat first finds overlaps between the reads. It then identifies sets of overlapping reads and constructs a variation graph from them. Finally the reads are corrected by aligning them to the graph. VeChat proceeds in two iterations with the same overall structure but subtle differences in the details.

The idea of building a graph based representation of reads and then aligning reads to it to correct them has been presented before for hybrid error correction (see LoRDEC, Jabba and GraphAligner for examples). These approaches use DBGs instead of variation graphs to

represent the reads but the DBG will include the variation present in the reads. However, I am not aware that this approach would have been applied to self-correction successfully and the hybrid methods were not particularly designed for preserving variation.

The novelty of this paper is to devise a method for building a variation graph based on the overlaps of the reads and to show that the approach works in self-correction and preserves variation.

The paper is well written. I was able to install the software and run a small example provided in the repository.

Suggestions for revision:

=====

Major

Comment: 1) The authors claim in the abstract and introduction that the current error correction methods all are based on raising a consensus sequence which reflects the summary of the observed reads. I find this a bit misleading as several hybrid methods such as LoRDEC, Jabba, and GraphAligner in fact represent the short reads with a de Bruijn graph and correct the long reads by aligning them to the de Bruijn graph. It is true that these methods were not designed specifically for preserving variation or to preserve low-frequency haplotypes/strains but still the approach is quite similar to the one presented in this paper.

Response: Thanks for pointing this out, we agree with that. In fact, the misunderstanding is due to us not specifying the consensus sequence claim in sufficient detail. While hybrid methods do make use of de Bruijn graphs, in a similar spirit as we do using variation graphs, there are no *self-correction* methods that do that. We have amended the corresponding sentences to account for that correctly:

- In the Abstract, we now say “*Current self-correction methods make use of a consensus sequence as a template.*” (instead of the earlier “*The current standard is to make use of a consensus sequence as a template.*”)
- In the Introduction, we now say “*The common denominator that unifies all of these self-correction approaches is to raise consensus sequence as a summary of the reads observed.*” (instead of the earlier “*The common denominator that unifies all of these approaches is to raise a consensus sequence that reflects a summary of the reads observed.*”).

Comment: 2) Metagenome data, lines 198-205: I think the most crucial parameter for the success of error correction is the sequencing coverage of the lowest abundance strain. Thus I would like to see here what is the range of sequencing coverage over the different strains, i.e. what is the sequencing coverage of the minimum abundance strain / the maximum abundance strain?

Response: Thank you for pointing this out, we agree that these are relevant quantities. We are now displaying the sequencing coverage information for each strain in the Supplementary Table S1. And we have added the following text at the end of this ‘Metagenome data’ paragraph:

“The corresponding sequencing coverages over different strains range from 12.6x to 68.0x and from 11.0x to 126.9x, respectively; see Supplementary Table S1 for information on coverage of individual strains.”

Comment: 3) The definition of the Support of an edge is not clear to me. On lines 512-513, it is stated that Support of an edge agrees with the number of reads covering that edge. However, on lines 506-509 it is not clear to me if it is enough that the nodes v and w appear in that order in a read induced path or if the nodes v and w need to be adjacent in that path, i.e. the edge (v,w) belongs to the path. The definition of the weight of an edge (Equation 2) seems to indicate that the nodes v and w do not even need to appear in that order in a read induced path but it would be enough for them to belong to the path. If the adjacency of the nodes is not required, this would imply that erroneous edges inserted to the graph due to deletion errors in the reads will have high weights because most read induced paths will visit both end points of the edge although they do not include the edge itself.

Response: Thank you for accurate observation; our explanation was sloppy indeed. In fact, the adjacency of the nodes ‘is’ required. We have changed the earlier lines 506-509 (*“If two consecutive nodes v,w , as particular subsets of items, do not appear in sufficiently many baskets, that is, are not contained in that particular order in sufficiently many read induced paths RP , the corresponding edge $e=(v,w)$ is removed from the graph.”*) by the sentences:

“If two nodes v,w that are connected by an edge $e=(v,w)$, as particular subsets of items, do not appear in sufficiently many baskets, that is, the corresponding edge $e=(v,w)$ is not contained in sufficiently many read induced paths RP , the corresponding edge is removed from the graph.”

Also, we have corrected Equation 2:

$$w(e) = w(v_i, v_j) := \frac{1}{2} \sum_{r \in R(v_i) \cap R(v_j)} w(v_i) + w(v_j) \quad (2)$$

to:

For an edge $e = (v_i, v_j)$, let $R(v_i, v_j)$ be all reads that cover edge e . We further determine

$$w(e) = w(v_i, v_j) := \frac{1}{2} \sum_{r \in R(v_i, v_j)} w(v_i) + w(v_j) \quad (2)$$

Comment: 4) The authors write that the pre-corrected reads generated by way of cycle 1 still contain a small, but yet non-negligible amount of random errors. It would be interesting to see the benchmarking results for some data set for the pre-corrected reads. This would help to evaluate what is the significance of cycle 2 in the method.

Response: We have Supplementary Table S15 to the supplement, see also below. The corresponding results show that cycle2 significantly improves the performance achieved by cycle1 alone. This justifies the claims raised in the manuscript. We are now also referencing Table S15 in subsection ‘Second Cycle: Modifications Step 1 and 4’ of the Methods section.

Method	#Reads	Error rate (%)	Mismatch (%)	Indel (%)	Haplotype coverage (%)	N50 (bp)	NGA50 (bp)	# Mis-assemblies
Ploidy=2								
Cycle 1	32244	0.206	0.032	0.174	100.0	12515	38441	1
Cycle 2	31958	0.014	0.006	0.008	100.0	12556	38515	0
Ploidy=3								
Cycle 1	49133	0.286	0.066	0.220	100.0	12515	38434	10
Cycle 2	48085	0.031	0.015	0.016	100.0	12595	38467	13
Ploidy=4								
Cycle 1	64539	0.317	0.097	0.220	99.9	12494	38432	47
Cycle 2	62743	0.074	0.047	0.027	99.9	12593	38442	44
Metagenome of low complexity (20 genomes)								
Cycle 1	299149	0.203	0.038	0.165	98.8	11816	29558	91
Cycle 2	293466	0.036	0.020	0.015	96.9	11866	29555	104

Table S15. Results of cycle 1 and cycle 2 in VeChat. Simulated PacBio CLR datasets of polyploid genomes (ploidy=2,3,4) and metagenome (low complexity) are shown.

Minor

Comment: 5) Page 11, line 322: An incomplete sentence: "Here, we report on quick application"

Response: It has been changed as: "Here, we report the quick application "VeChat + Assembler" on various polyploid genome and metagenome datasets."

Comment: 6) Page 17, line 570: "set of pre-corrected" -> "set of pre-corrected reads"

Response: We have fixed this.

Optional

Comment: 7) Section "Workflow", pages 3-5: Mentioning that the variation graph is constructed with the POA algorithm already in this section would help readers who are familiar with the methods in this area to quickly grasp the ideas. In the current manuscript, this of course becomes clear in the Methods section.

Response: As suggested, we have added this information (variation graph is constructed with the POA algorithm) in the Workflow section:

"Step 4 involves the construction of a variation graph using the partial order alignment (POA) algorithm (Lee et al., 2002), and pruning this graph in an iterative manner....."

REVIEWERS' COMMENTS

Reviewer #1 (Remarks to the Author):

Great work. The authors have addressed all the issues I raised and have improved the manuscript.

Reviewer #2 (Remarks to the Author):

All my concerns have been addressed.

Signed,

Leena Salmela